# Expanding or shrinking? range shifts in wild ungulates under climate change in Pamir-Karakoram mountains, Pakistan

**Hussain Ali[1], Jaffar Ud Din[2], Luciano Bosso[3], Shoaib Hameed[1], Muhammad Kabir[1], Muhammad Younas[2], Muhammad Ali Nawaz[4]***

**1** Department of Zoology, Quaid-I-Azam University, Islamabad, Pakistan, **2** Snow Leopard Trust, Pakistan Program, Islamabad, Pakistan, **3** Wildlife Research Unit, Dipartimento di Agraria, Università degli Studi di Napoli Federico II, Portici, Italy, **4** Environmental Science Program, Department of Biological and Environmental Sciences, Qatar University, Doha, Qatar

* nawazma@gmail.com

**Data Availability Statement:** All relevant data are within the manuscript and its Supporting information files.

## Abstract

Climate change is expected to impact a large number of organisms in many ecosystems, including several threatened mammals. A better understanding of climate impacts on species can make conservation efforts more effective. The Himalayan ibex (*Capra ibex sibirica*) and blue sheep (*Pseudois nayaur*) are economically important wild ungulates in northern Pakistan because they are sought-after hunting trophies. However, both species are threatened due to several human-induced factors, and these factors are expected to aggravate under changing climate in the High Himalayas. In this study, we investigated populations of ibex and blue sheep in the Pamir-Karakoram mountains in order to (i) update and validate their geographical distributions through empirical data; (ii) understand range shifts under climate change scenarios; and (iii) predict future habitats to aid long-term conservation planning. Presence records of target species were collected through camera trapping and sightings in the field. We constructed Maximum Entropy (MaxEnt) model on presence record and six key climatic variables to predict the current and future distributions of ibex and blue sheep. Two representative concentration pathways (4.5 and 8.5) and two-time projections (2050 and 2070) were used for future range predictions. Our results indicated that ca. 37% and 9% of the total study area (Gilgit-Baltistan) was suitable under current climatic conditions for Himalayan ibex and blue sheep, respectively. Annual mean precipitation was a key determinant of suitable habitat for both ungulate species. Under changing climate scenarios, both species will lose a significant part of their habitats, particularly in the Himalayan and Hindu Kush ranges. The Pamir-Karakoram ranges will serve as climate refugia for both species. This area shall remain focus of future conservation efforts to protect Pakistan's mountain ungulates.

**Funding:** This study received support from the following sources: Snow Leopard Trust (awarded to MAN); and Pakistan Snow Leopard and Ecosystem Protection Program (grant number 9231, awarded to MAN). The funders had no role in study design, data collection and analysis, decision to publish, or preparation of the manuscript.

**Competing interests:** The authors have declared that no competing interests exist.

## Introduction

Climate change has impacted ecosystems in unprecedented ways globally [1, 2], and appears to be unrelenting. These impacts are further complicated by rapid economic growth [3] and increasing human populations, especially in developing countries [4, 5].

Pakistan is a developing country and ranks as the seventh most vulnerable country to climate change [6]. Extreme temperatures, heavy rainfall, and floods are devastating several ecosystems in the country [7, 8]. Climate change impacts are most frequent in Pakistan's northern mountain ranges, including the Pamir-Karakoram, Himalayas, and Hindu Kush [9] where increasing temperatures, changes in cropping season, receding glaciers or outbursts, and heavy flooding [10–15] are leading to the extinction of several plant and animal species [16, 17]. These mighty mountains are a source of fresh water for half of South Asia [18, 19] and home to many floral and faunal species [20]. Furthermore, the Himalayas and Hindu Kush act as a barrier to monsoon rains [21] which helps the Karakoram range maintain its aridity. Highest and steepest among other ranges, the Karakoram is expected to be the one which is least affected by climate change [22].

Several species of wild ungulate, including the markhor (*Capra facolneri facolneri*), Ladakh urial (*Ovis vignei vignei*), Marco Polo sheep (*Ovis ammon polii*), Kashmir musk deer (*Moschus cupreus*), Himalayan ibex (*Capra ibex sibirica*), and blue sheep (*Pseudois nayaur*) live in these mountains. They play an important role in sustaining mountain ecosystems by influencing vegetation structure, plant composition, and nutrient recycling, in addition to being prey for carnivores [23]. However, climatic variations in recent years have impacted many ungulate species [3], and such impacts could have devasting effects on the ecosystem, including the carnivore community [24]. Climate studies in the Himalayas [25], western Tian Shan and Kyrgyz Alatau mountain ranges in Kazakhstan [26], Ghats in India [27], and Tibetan plateau in China [28] report climate change to be a serious threat to wild ungulates, leading to many species' extinction [3, 25, 27].

The Himalayan ibex is the most common of six wild ungulates in Pakistan. Its range historically extended from Swat to Khunjerab, although it has shrunk to the extreme northern parts of the country [29]. It is found in relatively arid precipitous mountain ranges living well above the tree line at elevations of 3,500–5,000 m [30]. The species does not enter forest zones, preferring steep escape terrain [31]. On the other hand, the blue sheep or *bharal* [32], an intermediate species between the goat and sheep [33] is found in less precipitous areas compared with ibex, at altitudes of 3,500–5,500 m in slopes covered with grasses and sedges, preferably with a southern-east exposition [34, 35].

The persistence of mountain ungulates like the Himalayan ibex and blue sheep in northern Pakistan is important because they are coveted trophies for hunters whose license fees help impoverished communities, who, in turn, help conserve biodiversity in far-flung areas [32]. Conservation planning that targets the long-term survival of these species is not only important from a nature perspective but is also vital for local human populations. Such planning must be informed by both current occurrence and future distribution of these iconic species in response to climate change. Currently, wild ungulate distributions in Gilgit-Baltistan (GB) is only partially known, and knowledge of climate change-induced impacts on species and habitats is insufficient [9]. We considered the ibex and blue sheep as model species to understand range shifts and other associated impacts of climate change on wild ungulates. The selected species represent two different groups—goats and sheep—and distinctive habitats. Inferences drawn from this study will, therefore, build knowledge for the informed management of wild ungulates in northern Pakistan. To achieve this objective, we used species distribution models

(SDMs) which are widely adopted in investigations of species distribution and range shifts [36, 37].

## Materials and methods

### Study area

Our study was conducted in Gilgit-Baltistan, Pakistan that lies between latitudes 36˚ N to 37˚ N and longitudes 74˚ E to 76˚ E, with an area ca. 72,200 km$^2$, dominated by glaciers and the snow-capped mountains of the Karakoram, Himalaya, Hindu Kush, and Pamir [38, 39]. The area is characterized by a variety of climatic conditions ranging from the monsoon-influenced moist temperate zone in the western Himalayas to the semi-arid cold deserts of the northern Karakorum and Hindu Kush [38]. There are numerous (forest) plant species, including the deodar *(Cedrus deodara)*, blue pine *(Pinus wallichiana)*, fir *(Abies spectabilis)*, spruce *(Picea smithina)*, chilgoza *(Pinus gerardiana)*, juniper *(Juniperus spp.)*, and birch *(Betula utilis)* [40], and 54 mammalian species [41], including rare ones [30] like the snow leopard *(Panthera uncia)*, Astor markhor *(Capra falconeri falconeri)*, Ladakh urial *(Ovis vignei vignei)*, Marco Polo sheep *(Ovis ammon polii)*, grey wolf *(Canis lupus)*, Himalayan lynx *(Lynx lynx)*, brown bear *(Ursus arctos)*, and musk deer *(Moschus spp.)*, in addition to the previously mentioned Himalayan ibex and blue sheep.

### Collection of presence records

Himalayan ibex and blue sheep presence records (Fig 1) were collected using two methods: camera trapping and double observer surveys.

1. Camera trapping: We installed 225 (Reconyx HC 500 and HC 900; Reconyx, Holmen, USA) cameras during the period 2010–2016 for *C. ibex sibirica* and *P. nayaur*, in different months of the year i.e., Khunjerab National Park (KNP) (November to January, 2010 and September to November, 2011), in Qurumber National Park (QNP) (May to June 2012) in Misgar Valley (May to July, 2013), in Hopper and Hisper Valleys (March to May, 2016) Cameras were left operational for 10 days in the first camera trapping in KNP, but in the latter surveys they were left operational for 40 days to increase the capture rate [42, 43].

2. Double observer Survey: We carried out this survey in 2012–2016 in different parts (KNP, Gojal Valley, Shigar Valley, in Skardu district, and in Gilgit district) of the study area by dividing it into smaller blocks based on watersheds. These watersheds were not larger than daily ungulate/human movement ability. Two observers were sent for survey separated by time (15 minutes) if only one trail was available, or by space, if two trails were available. Each watershed was surveyed by walking along pre-determined routes [44]. The locations where Himalayan ibex and blue sheep were sighted, have been used as presence points to build the MaxEnt model.

We collected 143 and 60 presence points for Himalayan ibex and blue sheep, respectively (S1A and S1B Fig). We then screened these presence points in ArcGIS 10.7 (ESRI, Redland, USA) using nearest neighbor analysis to check spatial autocorrelation [36, 43, 45]. This analysis revealed a high clustering among presence points. Aggregation was, therefore, spatially filtered using SDMTools [46] to ensure independence [36, 43, 47]. This operation led to 36 and 29 presence points for Himalayan ibex and blue sheep, respectively, which we used in MaxEnt models (Fig 1).

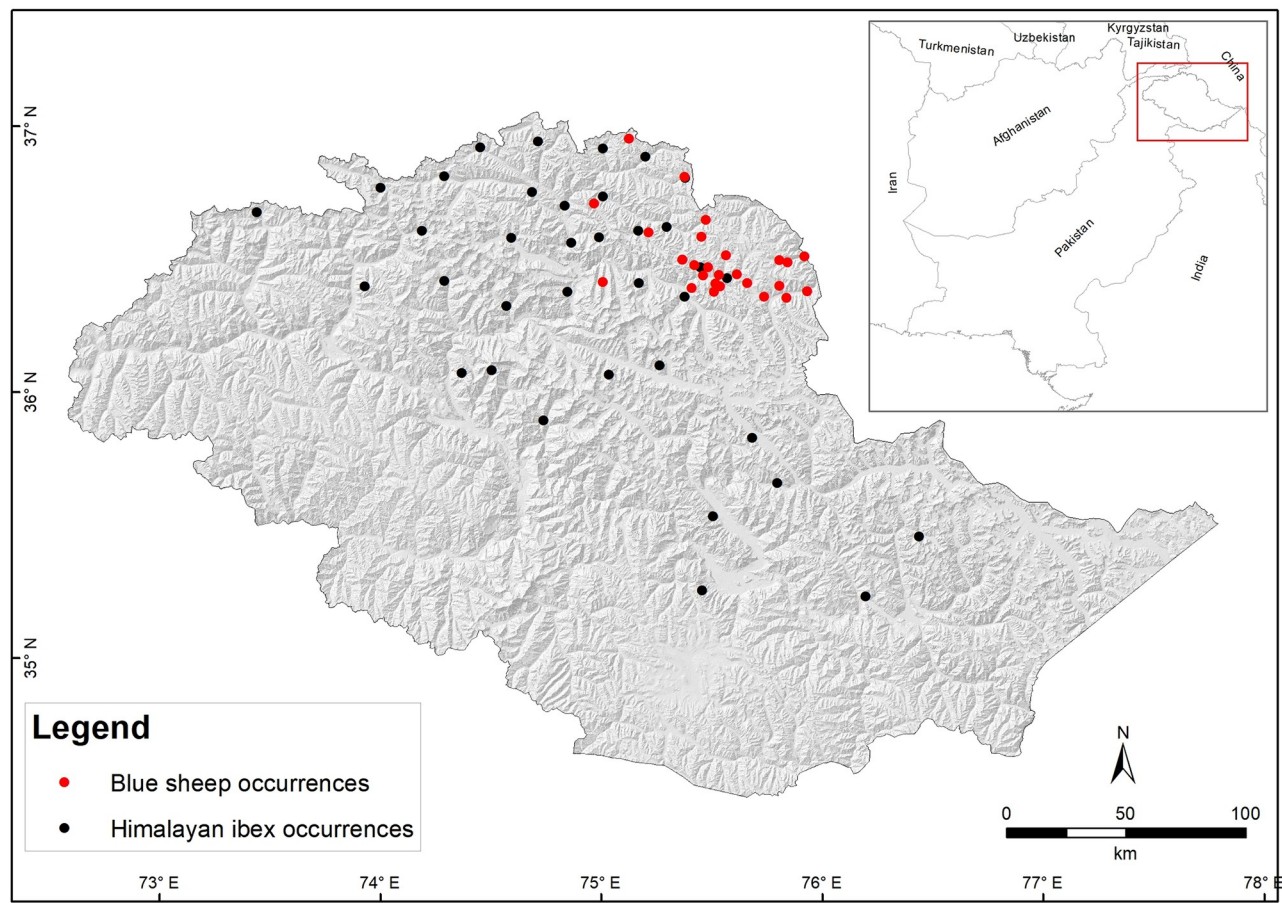

**Fig 1. Sampling locations of Himalayan ibex and blue sheep in GB, Pakistan used to build model.**

## Climatic variables

We downloaded 19 climatic variables from WorldClim 1.4 (https://www.worldclim.org/current) [48] to predict currently suitable areas for Himalayan ibex and blue sheep. All the variables were in raster files (grid) with 30-arc second resolution ($0.93 \times 0.93$ km = $0.86$ km$^2$ at the equator). Further details and information on the realization and interpretation of the WorldClim variables used in this study can be found at https://pubs.usgs.gov/ds/691/. We checked all variables for multicollinearity and excluded highly correlated variables i.e., r $\geq 0.70$ (Pearson's correlation coefficient) [43]. This process led to use in the modeling analysis of six environmental variables: annual mean temperature (C˚), mean diurnal range (˚C), temperature seasonality [(standard deviation * 100) (˚C)], mean temperature of wettest quarter (˚C), mean precipitation (mm), and precipitation seasonality (%).

We used global circulation models (GCMs) MIROC5, BCC-CSM1-1, CCSM4, and Had-GEM2ES to predict the future distribution of Himalayan ibex and blue sheep under climate change conditions. Various organizations developed these models under the Coupled Model Intercomparison Project, phase 5 (CMIP5) and are considered highly reliable [36, 49]. The future projections of these GCMs are based on representative concentration pathways (RCPs) which are greenhouse gas (GHG) concentration trajectories on a range of radiative forces suggested in the Intergovernmental Panel on Climate Change's (IPCC) fifth assessment report [50]. We used RCP 4.5 and RCP 8.5 the former is a moderate GHG mitigation scenario [51]

where emissions will peak around 2040 and then decline, while the latter is a scenario where GHG emissions will be the highest of all four RCPs (2.6, 4.5. 6.0 and 8.5) throughout the 21st century [27].

## Modeling procedure

We used MaxEnt ver. 3.4.1 [52] to predict the current and future distribution of *C. ibex sibirica* and *P. nayaur* in Pakistan [25]. MaxEnt is a piece of machine learning software used to develop SDMs [53–55]. It is capable of predicting species distribution using presence-only data [56] and predicting the distribution of poorly known species [36, 57]. We built the model using a logistic output format to yield environmental suitability ranging from 0 (unsuitable) to 1 (highly suitable) [58]. We fixed the regularization multiplier to 1, selected 5,000 iterations [27], and ran 20 replicates with cross-validations tests [43].

Different GCM projections can have inherited uncertainties. To avoid this, we used area under the curve (AUC) scores as weighting coefficients that resulted from 20 cross-validations for each of four GCMs and produced a single forecast for each time scale by averaging all individual GCMs for that time slice. [28, 59–61]. We used ten percentile training presence values as the threshold to develop binary presence/absence maps [43].

The model was projected to entire GB. To project the models calibrated for survey area over entire GB, the variables in the projection area must meet a condition of environmental similarity with the environmental data used for calibrating the model. Therefore, we preliminarily ascertained that this condition was verified for both current and future projections by inspecting Multivariate Environmental Similarity Surfaces (MESS), the MESS calculates the similarity of each point in the region of projection to a set of reference points (e.g., background data) and maps the results [56] MESS maps produced by MaxEnt can help users identify extrapolated areas and provide a quantitative measure of projection uncertainty.

## Model validation

We tested the predictive performance of the models with different methods: receiver operated characteristics, analyzing the AUC [62], and the true skill statistic (TSS) [63]. AUC assesses models' discrimination ability with values ranging from 0 (equaling random distribution) to 1 (perfect prediction). TSS compares the number of correct forecasts minus those attributable to random guessing, to that of a hypothetical set of perfect forecasts. It considers both omission and commission errors and success as a result of random guessing. Its values range from -1 (a performance no better than random) to +1 (perfect agreement).

## Niche overlap

We calculated the niche overlap between *C. ibex sibirica* and *P. nayaur* for predicted habitats using ENMTools [64] in the current time and future climate change scenarios. ENMTools uses MaxEnt map values of habitat suitability for each grid and measures niche overlap using D and I values [64]. It uses Schoener's D value to calculate niche overlap and gives probability distributions with values ranging from 0 (no overlap) to 1 (complete overlap). Similarly, Hellinger's I-statistic in ENMTools measures models' ability to estimate true suitability [64].

## Results

### Model performance

The AUC values for our models were 0.969 ± 0.025 and 0.821 ± 0.138 for blue sheep and Himalayan ibex, respectively. TSS values were 0.841 ± 0.007 and 0.454 ± 0.281 for blue

sheep and Himalayan ibex, respectively. Both tests suggest strong performances of our models.

## Current distribution of Himalayan ibex and blue sheep

Our binary maps showed ca. 26 500 km$^2$ (37.71% of total study area) and ca. 6 500 km$^2$ (9.26% of total study area) suitable for Himalayan ibex and blue sheep, respectively (Fig 2).

We found that the current habitat predicted for Himalayan ibex included the latitudes from 34˚ to 37˚ and the longitudes from 73˚ to 77˚. The most suitable habitats fell in the Karakoram range, followed by the Hindu Kush, and then to a minor extent in the Himalayas (Fig 2A). The habitat suitability of Himalayan ibex was predicted in all ten districts of GB with strongholds in Hunza, Nagar, Shigar, and Ghanche districts. We found that habitats suitable to blue sheep were between the latitudes 35˚ to 37˚ and the longitudes 74˚ to 77˚ along the Pakistan-China border in the Pamir-Karakorum range that administratively falls in Hunza district, followed by some parts of the Shigar and Ghanche districts along the Pakistan-China border (Fig 2B). We found that annual mean precipitation, mean temperature of the wettest quarter, and temperature seasonality were the most important variables (with 91.6% contribution) in predicting suitable habitats for blue sheep (S1 Table). For ibex, annual mean precipitation, annual mean temperature, and precipitation seasonality were key habitat predictors with an 89% contribution (S2 Table).

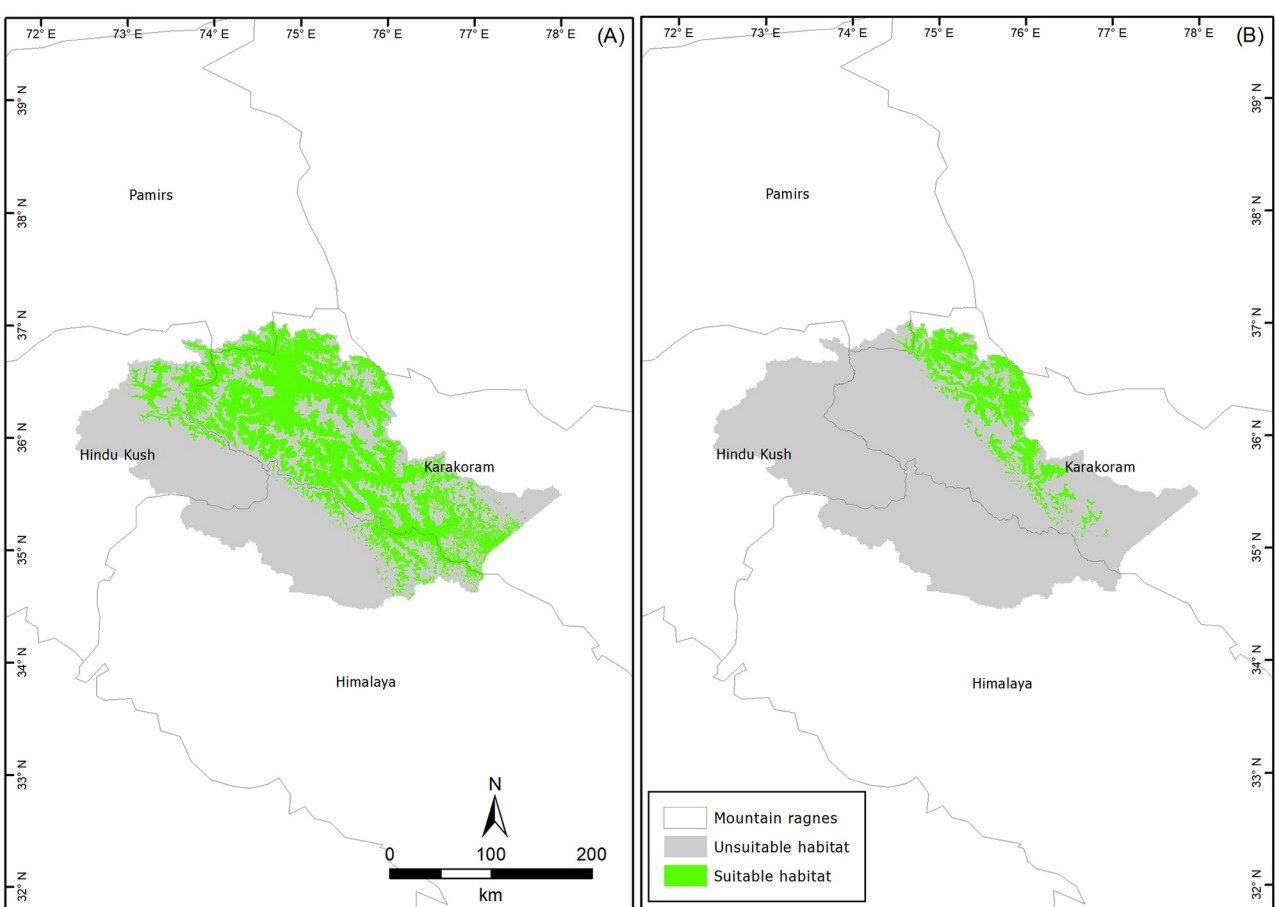

**Fig 2. Binary maps of habitat suitability for Himalayan ibex (A) and blue sheep (B) under current climatic conditions.**

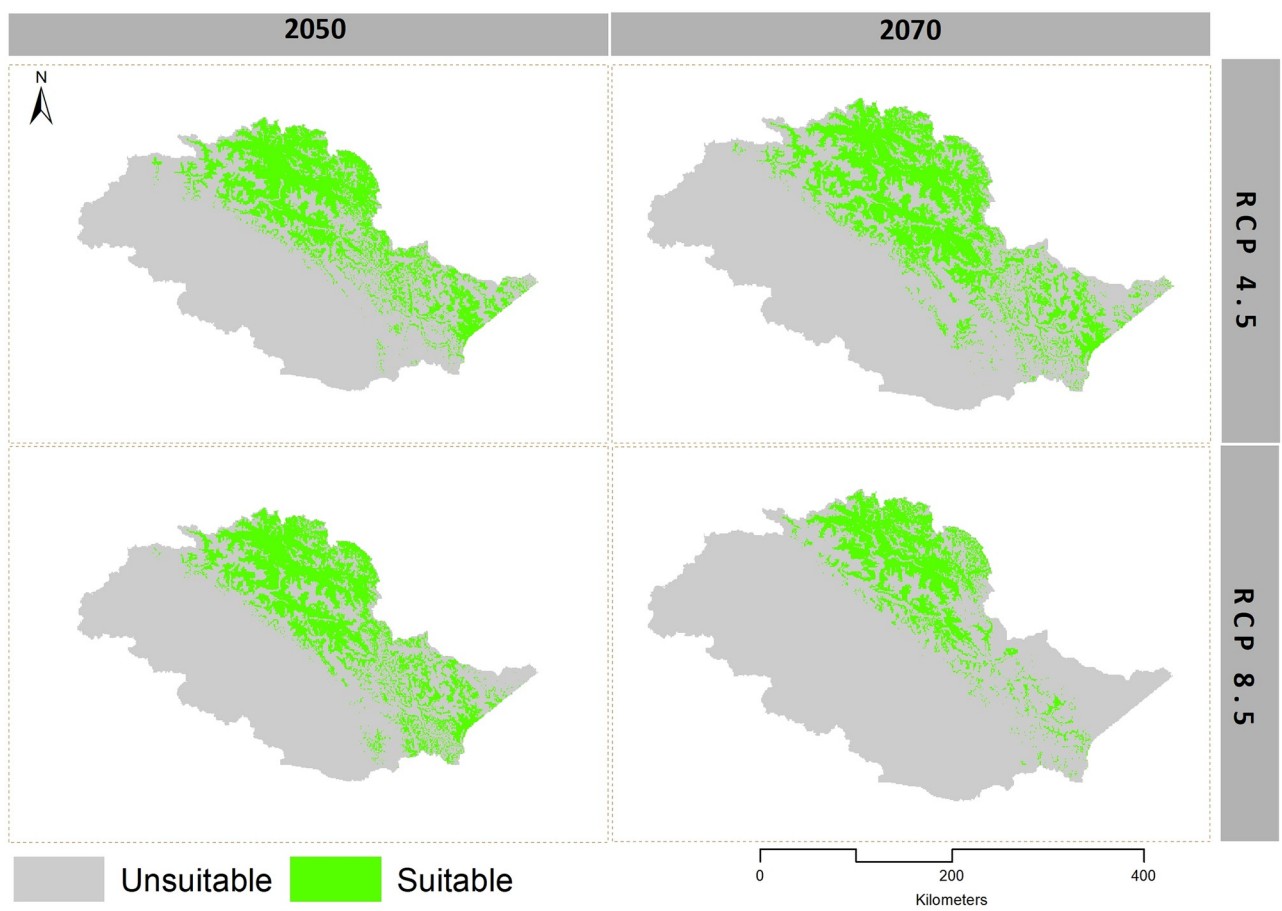

**Fig 3. Binary maps of Himalayan ibex under RCP 4.5 and RCP 8.5 scenarios in 2050 and 2070.**

### Future distribution of Himalayan ibex and blue sheep

Our models showed habitat shrinkage for both Himalayan ibex and blue sheep for RCP 4.5 and RCP 8.5, in 2050 and 2070 scenarios (Figs 3 and 4, Tables 1 and 2).

In the extreme climate change scenario (RCP 8.5 of 2070), blue sheep lost (58%) from the suitable areas that it has currently occupied and gained new suitable areas by extending its current range towards the east. Himalayan ibex gained the least and lost (64.80%) in RCP 8.5 of 2070 (Table 3 and Figs 5 and 6). The model predicted habitat shrinkage to an area of 2,515 km$^2$ for blue sheep and 9,248 km$^2$ for ibex under the extreme climate change scenario.

The center of suitable Himalayan ibex habitat gradually shifted from the north to the east in RCP 4.5 and RCP 8.5 of 2050, and RCP 4.5 of 2070, while in RCP 8.5 of 2070, it again shifted from the east to the north. The center of the suitable habitat of blue sheep first shifted gradually from the west towards the north in RCP 4.5 and RCP 8.5 of 2050, and RCP 4.5 of 2070. In RCP 8.5 of 2070, it shifted towards the east from the north. The MESS analysis predicted some areas with novel climate conditions across the range for both *P. nayaur* and *C. ibex sibirica* in the future projections. However, these areas were found outside the training range of our model (S1–S8 Figs).

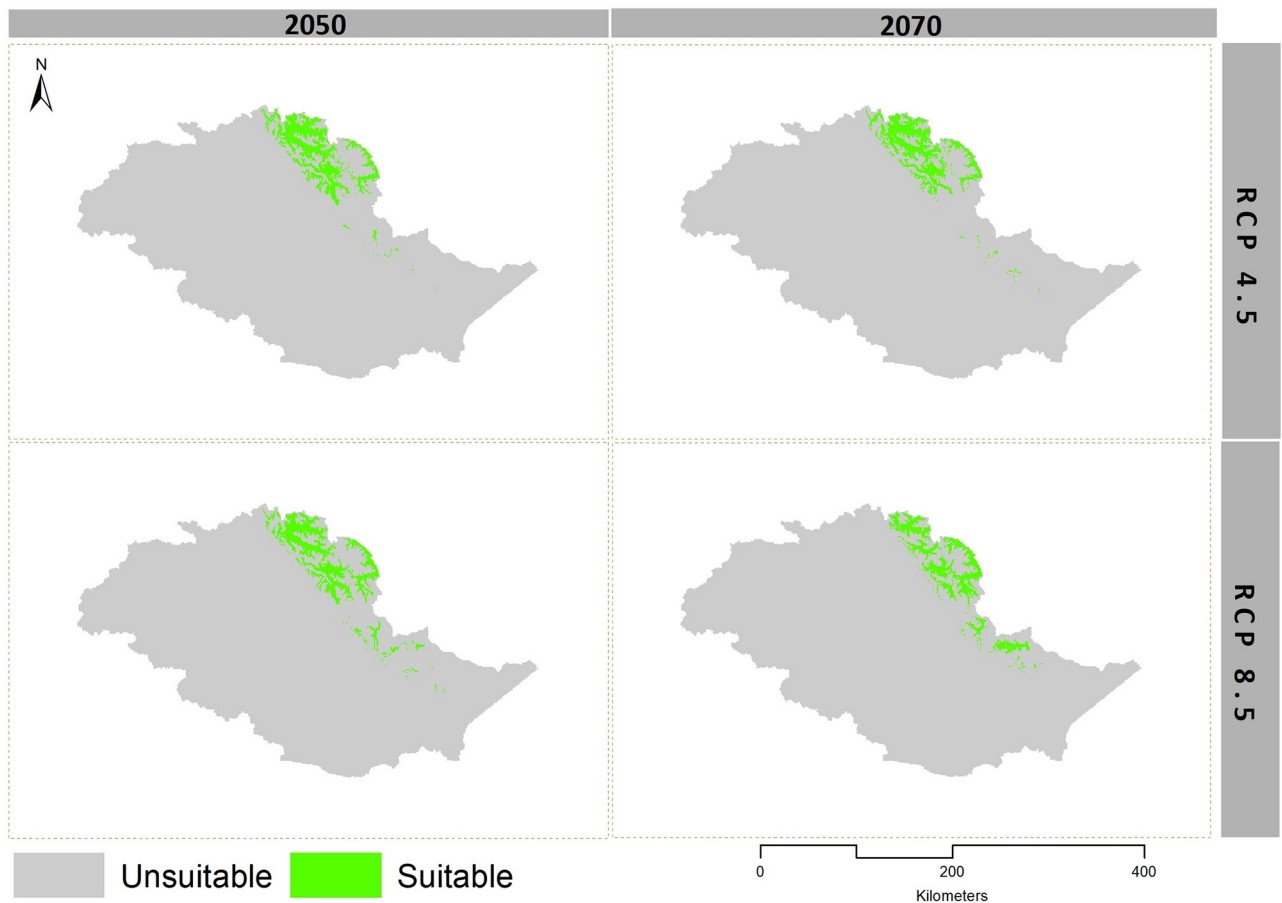

**Fig 4. Binary maps of blue sheep under RCP 4.5 and RCP 8.5 scenarios in 2050 and 2070.**

## Niche overlap

Our analysis of niche overlap between blue sheep and Himalayan ibex indicated a moderate level of niche overlap in the current time. ANOVA test showed that the mean of Schoener's D value for two climate change scenarios (4.5 and 8.5) did not vary significantly ($F_{(3,12)} = 0.15$, $p = 0.68$) on the temporal scale (2050 vs. 2070). Similarly, the probability-based I-statistic values for niche overlap were also not significantly different ($F_{(3, 12)} = 0.37$, $p = 77$) for different RCPs of different years (Table 4 and Fig 7).

**Table 1. Area predicted to be suitable in the current and different future climate change scenarios within GB for blue sheep.**

|   | Scenario | No. of pixels predicted to be suitable | Percentage reduction in future scenarios |
|---|---|---|---|
| 1 | Current | 9,035 | - |
| 2 | 2050 RCP 4.5 | 3,922 | 56.59 |
| 3 | 2050 RCP 8.5 | 4,039 | 55.29 |
| 4 | 2070 RCP 4.5 | 3,738 | 58.62 |
| 5 | 2070 RCP 8.5 | 3,491 | 61.93 |

**Table 2. Area predicted to be suitable in the current and different future climate change scenarios within GB for *C. ibex sibirica*.**

| | Scenario | No. of pixels predicted to be suitable | Percentage reduction in future scenarios |
|---|---|---|---|
| 1 | Current | 36,790 | - |
| 2 | 2050 RCP 4.5 | 23,797 | 35.31 |
| 3 | 2050 RCP 8.5 | 23,804 | 35.29 |
| 4 | 2070 RCP 4.5 | 24,391 | 33.70 |
| 5 | 2070 RCP 8.5 | 12,950 | 64.80 |

**Table 3. Change resulting from climate change in suitable habitats of blue sheep and Himalayan ibex.**

| Species | Future | Scenario | Expansion | No occupancy | Stable areas | Habitat loss | Total |
|---|---|---|---|---|---|---|---|
| Blue sheep | 2050 | RCP 4.5 | 3.60 | 63,779 | 2,822 | 3,687 | 70,291 |
| | 2050 | RCP 8.5 | 47.55 | 63,735 | 2,906 | 3,604 | 70,292 |
| | 2070 | RCP 4.5 | 23.05 | 63,759 | 2,670 | 3,839 | 70,291 |
| | 2070 | RCP 8.5 | 125.38 | 63,657 | 2,390 | 4,120 | 70,292 |
| Himalayan ibex | 2050 | RCP 4.5 | 3,024 | 40,738 | 14,126 | 12,460 | 70,348 |
| | 2050 | RCP 8.5 | 2,957 | 40,805 | 14,175 | 12,411 | 70,348 |
| | 2070 | RCP 4.5 | 3,363 | 40,009 | 14,102 | 12,330 | 69,804 |
| | 2070 | RCP 8.5 | 1,035 | 42,228 | 8,213 | 18,255 | 69,731 |

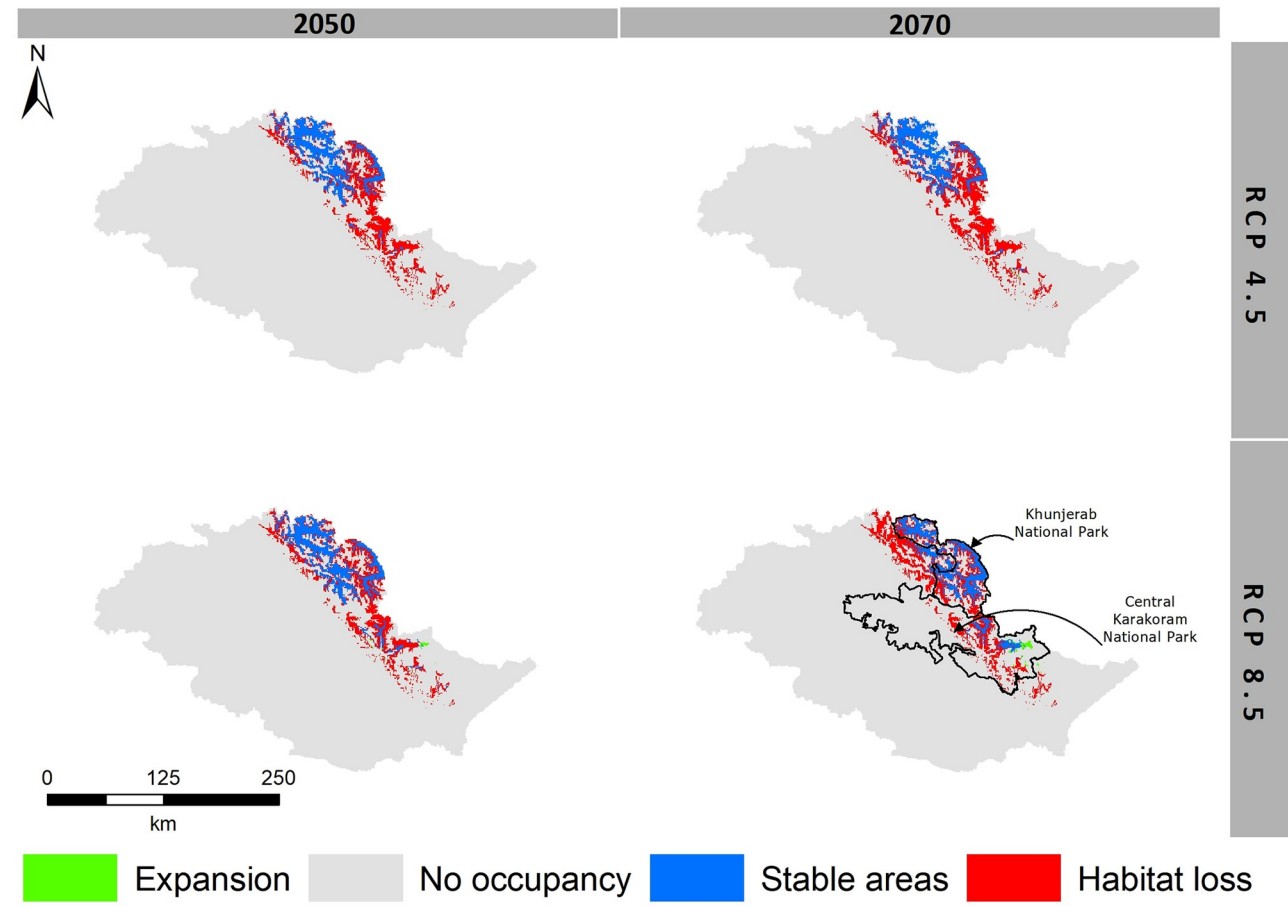

**Fig 5. The predicted change in the suitable habitats of blue sheep in 2050 and 2070 under RCP 4.5 and RCP 8.5 scenarios.**

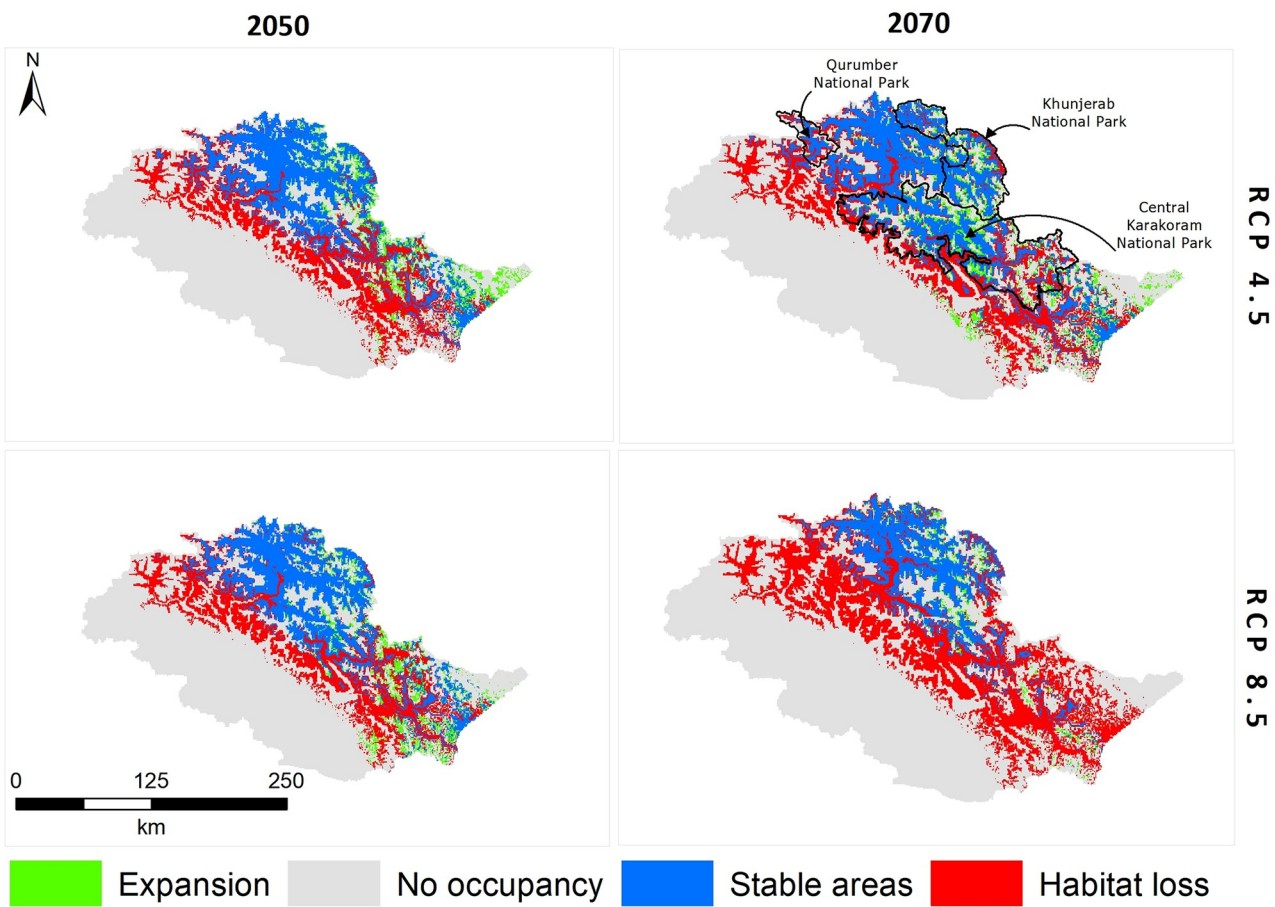

**Fig 6. The predicted change in the suitable habitats of blue sheep in 2050 and 2070 under RCP 4.5 and RCP 8.5 scenarios.**

## Discussion

The use of SDMs for the predictive distribution of biodiversity [65] has increased as the approach is considered efficient in predicting species distribution and climate change impact [66] which aids in species conservation planning [55]. MaxEnt is widely used for its proven ability to construct models using presence-only data [67]. This model worked well on our presence data as indicated by the AUC scores (>0.8), which places it among the best-published models [25, 26, 28, 68]. The higher TSS values further supported the credibility of results [36, 47].

The melting of Himalayan glaciers has increased in the 21st century [69] while the glaciers of the Hindu Kush and Karakoram will melt at a slower rate [70]. In fact, some glaciers in the

**Table 4. Estimation of niche overlap between Himalayan ibex and blue sheep under different climate change scenarios.**

| Schoener's niche overlap metric | Current | 2050 | | 2070 | |
|---|---|---|---|---|---|
| | | RCP 4.5 | RCP 8.5 | RCP 4.5 | RCP 8.5 |
| D | 0.42 | 0.44 | 0.46 | 0.44 | 0.47 |
| I | 0.69 | 0.72 | 0.73 | 0.72 | 0.74 |

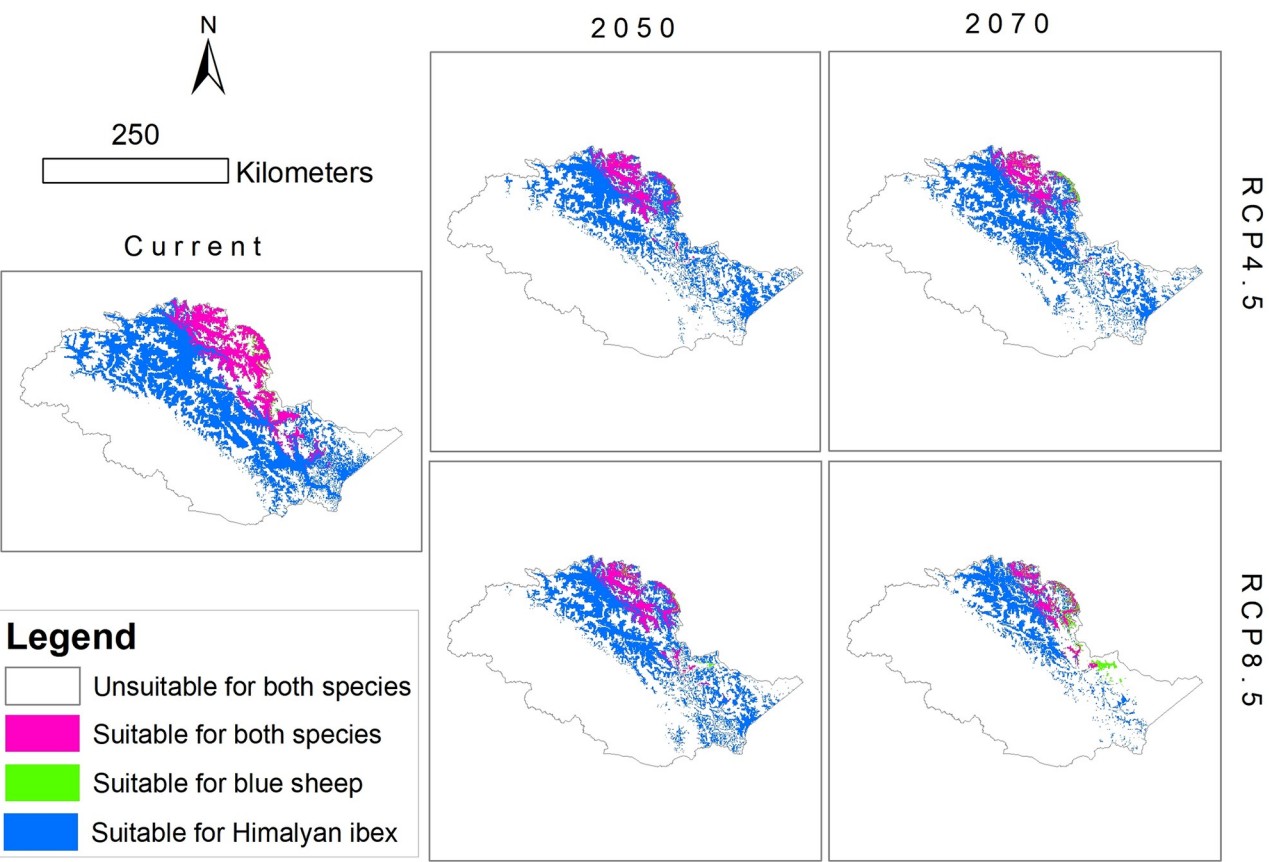

**Fig 7. The spatial pattern of niche overlap between blue sheep and Himalayan ibex in current and different climate change scenarios.**

higher watersheds of the Karakoram are expanding [71] although at the same time they are thinning. However, regardless of the three described scenarios, the snow on these glaciers regulates ecological processes and patterns [72] and any change in glacier mass, negative or positive, will affect associated biodiversity. Our results for habitat loss and gain were strikingly aligned with the existing knowledge on glaciology. We found that global climate change will have significant effects on the habitats of mountain ungulates in northern Pakistan, though these effects are more pronounced in Hindu Kush, and Himalaya ranges.

Our model for current time predicted 6,510 km$^2$ and 26,510 km$^2$ of suitable area for blue sheep and Himalayan ibex, respectively. Both model species are present in most of the predicted habitats, or they occupied those areas historically [30, 33]. Ironically, Khan et al., (2014) reported sighting records of ibex in Tangir Valley of Diamer district, which is beyond the suitable habitat predicted in the current study, as well as outside of the former IUCN range [73]. This probably indicates southwards expansion of ibex in recent years. Our model predicted suitable habitat for blue sheep on the Braldu glacier where sheep do not currently exist [74]. Interestingly, older records indicate the presence of blue sheep in this area, e.g., [29] quote a sighting by T. J. Roberts in this area in 1975.

Both blue sheep and Himalayan ibex habitats are usually between the timber and snow lines at elevations of 3,500–5,500 m, and differ as Blue sheep prefers habitats with steep rolling hills and Himalayan ibex prefer precipitous habitats [33]. These habitats are usually

devoid of thick vegetation. Hence, precipitation is a vital factor to sustain life in this zone. We found annual precipitation to be the most contributing variable in predicting suitable habitat for both blue sheep and Himalayan ibex. Annual mean temperature was the second most important variable for Himalayan ibex, and temperature of wettest quarter the second most important for blue sheep. The dry habitats of both ibex and blue sheep have short growing seasons, and any weather fluctuation might leave species starving [75]. *Artemisia* and *Ephedra* shrubs are described as the ibex's main food sources [33]. A year of good winter precipitation and normal mean summer temperature enables shrubs to maximize their growth and green cover [76]. Blue sheep's preferred diet comprises of grasses, forbs, and shrubs *Berberis*, *Polygonum*, and *Ephedra*, respectively [33]. Even in the summers, precipitation at elevations above 4,000 m can bring temperatures below zero and constraint vegetative growth [76]. Hence, temperatures of wettest quarters (June, July, and August) play a decisive role in selecting suitable habitat for blue sheep. Khan et al. (2016) found annual precipitation and minimum temperature to be important variables for developing suitability models for *C. ibex sibirica* and *P. nayaur*, respectively. Aryal et al. (2016) and Luo et al. (2015) reported annual mean temperature as the most influencing variable in predicting suitable habitat for *P. nayaur*.

We observed a sharp decline (56% in RCP 4.5 and 58% in RCP 8.5) in the currently available suitable habitat for blue sheep and (33.70% in RCP 4.5 and 64.80% in RCP 8.5) for Himalayan ibex in extreme climate change scenarios for 2070. This is consistent with [25]who observed a decrease in blue sheep suitable habitat in the future due to climate change in Nepal. Similarly, Luo et al. (2015) reported a 30–50% range reduction for ungulates on the Tibetan plateau under different climate change scenarios.

Climate drives evolutionary processes, forcing animals to migrate to higher elevations or extend their distributional ranges towards the Northern Hemisphere [77] or eastward direction [28]. This process is believed to have occurred in the Miocene Epoch when members of the *Caprinae* in Eurasia and Africa began inhabiting the newly formed mountain ranges of the Himalayas, Karakoram, Hindu Kush, and Pamirs, which emerged from the sea during the Tertiary Period [33]. We expect a similar migration in northern Pakistan because the centers of predicted suitable habitat for Himalayan ibex will shift from north to east in RCP 4.5 and RCP 8.5 of 2050 and 2070 and again from east to the north in RCP 8.5 of 2070. For Himalayan ibex, it will shift from west to north in RCP 4.5 and RCP 8.5 of 2050 and 2070 and from north to east in RCP 8.5 of 2070.

Species co-evolved over millions of years, enabling them to co-exist by selecting different niches [78]. Our model predicted a moderate niche overlap between blue sheep and Himalayan ibex, and this overlap was predicted to increase if the extreme climatic conditions assumed in future scenarios prevail. Increasing temperatures and precipitation have already impacted Himalayan flora [79]. Alpine habitats have short growing seasons [80, 81] and offer relatively few species of grasses, sedges, forbs, shrubs, ferns, lichens, and mosses to Himalayan ibex and blue sheep [82–84]. Hence, these climatic changes in alpine ranges will increase the chances of habitat mismatch for many floral species [28, 80]. Climate change, together with anthropogenic effects transforming land for agriculture or afforestation, road construction, and mining could further shrink habitats suitable for ungulates [28, 68], potentially affecting their perpetuity and the proper functioning of ecosystems [85, 86].

Conservationists emphasize on locating habitats likely to be least affected by climate change and continue serving as suitable habitats (future refugia), and protecting them from anthropogenic activities [21, 87, 88]. Our model predicted such climate refugia for Himalayan ibex to be comprised of three national parks: Khunjerab National Park (KNP), Central Karakoram National Park (CKNP), and Qurumbar National Park (QNP) (Fig 6). For blue sheep, such

refugia exists in the buffer zone of KNP, along with a few patches on the Braldu glacier of CKNP (Fig 5). It is noteworthy, however, that Himalayan ibex will lose most of its current suitable habitat in CKNP in Baltistan division and areas around QNP in the future, but the areas of CKNP in Nagar district will remain stable. All three mountain ranges in our study area provide vital habitats to several mountain ungulates. Unfortunately, most of suitable habitats in Hindu Kush and Himalayas are expected to be altered under future scenarios. On contrary, the Pamir-Karakoram is likely to remain stable and continue accommodating both Himalayan ibex and blue sheep. The relatively lower effect of climate change in this range is likely due to the barrier effect of the Hindu Kush and Himalayas which blunt the monsoon, helping maintain the aridity of the Karakorum's' alpine steppes [21, 71].

## Conclusions

Our study demonstrate that the current suitable habitat of Himalayan ibex and blue sheep are vulnerable to climate change. Under the rapid climate change Himalayan ibex will lose most of its current suitable habitat in Himalayans and Hindu Kush while blue sheep that currently exists only in Pamir-Karakoram range will be slightly affected. The current network of protected areas (KNP and CKNP) will serve climate refugia for mountain ungulates.

There is urgent need to revisit protected areas management strategies in Pakistan, to enhance their effectiveness for conservation of mountain ungulates. The finding of this study can be used to revisit or align boundaries of existing protected areas with the future predicted habitats. Management and protection efforts shall remain disproportionally higher in parks that encompass climate refugia for mountain ungulates of the region.

## Supporting information

**S1 Map. Map showing unfiltered and retained occurrences used for the current study A) Himalayan ibex (total 143 points, retained points 36) B) Blue sheep (total 60 points, retained points 29) using SDMtoolbox V1.1(Brown 2014).**
(DOCX)

**S1 Table. Estimates of relative contributions of the environmental variables used to build MaxEnt model for blue sheep.**
(DOCX)

**S2 Table. Estimates of relative contributions of the environmental variables used to build MaxEnt model for Himalayan ibex.**
(DOCX)

**S1 Fig. Maps illustrating multivariate environmental similarity surface (MESS) approach as described in (Elith et al. 2010) and the most dissimilar variables(MOD) for Himalayan ibex under the year 2050 Representative Concentration Pathway (RCP4.5) for different Global Circulation Models.** Negative values indicate novel climate in the MESS map across the range. b) Most dissimilar variables (MOD) analysis shows those novel climatic conditions and the associated variables.
(DOCX)

**S2 Fig. Maps illustrating multivariate environmental similarity surface (MESS) approach as described in(Elith et al. 2010) and the most dissimilar variables(MOD) for Himalayan ibex under the year 2050 Representative Concentration Pathway (RCP8.5) for different Global Circulation Models.** Negative values indicate novel climate in the MESS map across

the range. b) Most dissimilar variables (MOD) analysis shows those novel climatic conditions and the associated variables.
(DOCX)

**S3 Fig. Maps illustrating multivariate environmental similarity surface (MESS) approach as described in (Elith et al. 2010) and the most dissimilar variables(MOD) for Himalayan ibex under the year 2070 Representative Concentration Pathway (RCP4.5) for different Global Circulation Models.** Negative values indicate novel climate in the MESS map across the range. b) Most dissimilar variables (MOD) analysis shows those novel climatic conditions and the associated variables.
(DOCX)

**S4 Fig. Maps illustrating multivariate environmental similarity surface (MESS) approach as described in (Elith et al. 2010) and the most dissimilar variables(MOD) for Himalayan ibex under the year 2050 Representative Concentration Pathway (RCP4.5) for different Global Circulation Models.** Negative values indicate novel climate in the MESS map across the range. b) Most dissimilar variables (MOD) analysis shows those novel climatic conditions and the associated variables.
(DOCX)

**S5 Fig. Maps illustrating multivariate environmental similarity surface (MESS) approach as described in (Elith et al. 2010) and the most dissimilar variables(MOD) for Blue sheep under the year 2050 Representative Concentration Pathway (RCP4.5) for different Global Circulation Models.** Negative values indicate novel climate in the MESS map across the range. b) Most dissimilar variables (MOD) analysis shows those novel climatic conditions and the associated variables.
(DOCX)

**S6 Fig. Maps illustrating multivariate environmental similarity surface (MESS) approach as described in (Elith et al. 2010) and the most dissimilar variables(MOD) for Blue sheep under the year 2050 Representative Concentration Pathway (RCP8.5) for different Global Circulation Models.** Negative values indicate novel climate in the MESS map across the range. b) Most dissimilar variables (MOD) analysis shows those novel climatic conditions and the associated variables.
(DOCX)

**S7 Fig. Maps illustrating multivariate environmental similarity surface (MESS) approach as described in (Elith et al. 2010) and the most dissimilar variables(MOD) for Blue sheep under the year 2070 Representative Concentration Pathway (RCP4.5) for different Global Circulation Models.** Negative values indicate novel climate in the MESS map across the range. b) Most dissimilar variables (MOD) analysis shows those novel climatic conditions and the associated variables.
(DOCX)

**S8 Fig. Maps illustrating multivariate environmental similarity surface (MESS) approach as described in (Elith et al. 2010) and the most dissimilar variables(MOD) for Blue sheep under the year 2070 Representative Concentration Pathway (RCP8.5) for different Global Circulation Models.** Negative values indicate novel climate in the MESS map across the range. b) Most dissimilar variables (MOD) analysis shows those novel climatic conditions and the associated variables.
(DOCX)

## Acknowledgments

We acknowledge the support provided by the staff of Parks and Wildlife Department and communities during the field surveys.

## Author Contributions

**Conceptualization:** Hussain Ali, Muhammad Ali Nawaz.

**Data curation:** Hussain Ali, Jaffar Ud Din, Shoaib Hameed, Muhammad Kabir, Muhammad Younas, Muhammad Ali Nawaz.

**Formal analysis:** Hussain Ali, Luciano Bosso, Shoaib Hameed, Muhammad Ali Nawaz.

**Funding acquisition:** Muhammad Ali Nawaz.

**Investigation:** Jaffar Ud Din, Muhammad Kabir.

**Methodology:** Hussain Ali.

**Project administration:** Jaffar Ud Din, Muhammad Ali Nawaz.

**Supervision:** Muhammad Ali Nawaz.

**Visualization:** Hussain Ali.

**Writing – original draft:** Hussain Ali.

**Writing – review & editing:** Luciano Bosso, Muhammad Ali Nawaz.

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
