## [Decision Letter · Decision Letter 0]

28 Jul 2021

PONE-D-21-17916

Expanding or shrinking? range shifts in wild ungulates under climate change in Pamir-Karakoram Mountains, Pakistan

PLOS ONE

Dear Dr. Nawaz,

Thank you for submitting your manuscript to PLOS ONE. After careful consideration, we feel that it has merit but does not fully meet PLOS ONE’s publication criteria as it currently stands. Therefore, we invite you to submit a revised version of the manuscript that addresses the points raised during the review process.

We look forward to receiving your revised manuscript.

Kind regards,

Tzen-Yuh Chiang

Academic Editor

PLOS ONE

Journal Requirements:

https://journals.plos.org/plosone/s/file?id=ba62/PLOSOne_formatting_sample_title_authors_affiliations.pdf\\

3. We note that Figure(s) 1, 2, 3, 4, 5, 6, and 7 in your submission contain map images which may be copyrighted. All PLOS content is published under the Creative Commons Attribution License (CC BY 4.0), which means that the manuscript, images, and Supporting Information files will be freely available online, and any third party is permitted to access, download, copy, distribute, and use these materials in any way, even commercially, with proper attribution. For these reasons, we cannot publish previously copyrighted maps or satellite images created using proprietary data, such as Google software (Google Maps, Street View, and Earth). For more information, see our copyright guidelines: http://journals.plos.org/plosone/s/licenses-and-copyright.

1. You may seek permission from the original copyright holder of Figure(s) 1, 2, 3, 4, 5, 6, and 7 to publish the content specifically under the CC BY 4.0 license.  

Reviewers' comments:

Reviewer's Responses to Questions

**Comments to the Author**

1. Is the manuscript technically sound, and do the data support the conclusions?

Reviewer #1: Partly

2. Has the statistical analysis been performed appropriately and rigorously? 

Reviewer #1: I Don't Know

3. Have the authors made all data underlying the findings in their manuscript fully available?

Reviewer #1: Yes

4. Is the manuscript presented in an intelligible fashion and written in standard English?

Reviewer #1: Yes

5. Review Comments to the Author

Reviewer #1: In general, an interesting item and approach. Some information missing as regards the field methods used and the selected parameters with a predictive value. The number of figures should be reduced and figures should be enriched with boundaries (main parks, major mountain systems). Several inaccuracies found across MS, as follows:

Line 63: Moschus instead of Mouchus

Lines 75-76: please rephrase in something like: “It was found in relatively aris precipitous mountain ranges, living well above the tree line at elevations of 3500-5000 m.”

Lines 77-81: please rephrase in something like: “On the other hand, the blue sheep or bharal, an intermediate wild caprine species between the goat and the sheep, is found in less precipitous areas compared with ibex, at altitudes of 3500-5500 m in slopes covered with grasses and sedges.” Low (or gentle) slopes may be confusing

Lines 88-89: please rephrase in something like: “Currently, wild ungulate distribution in Gilgit-Baltistan (GB) is only partially known, and knowledge of climate-change induced impacts on species and habitats is insufficient.” The distribition of wild ungulates is locally very well known (e.g., within CKNP)

Line 110: use “rare” and delete “endangered”

Line 121: “in all potential areas” is a vague expression. Please be more accurate in describing the criteria underpinning the positioning of cameras

Line 122: substitute “installed” with “operated” or something similar

Line 122: add details on the season/s when cameras where positioned. In addition, explain the reasons for such a wide range (10-40) in camera operation days

Lines 123-128: please, help the reader understand the reason/s for use of a “double observer survey” approach for the purposes of this study

Line 129: what do Authors exactly mean as “records”? Please define

Lines 167-175: There is something missing in this phrase (like some “.”or “,”). Please review with accuracy.

Lines 213-214: Fig. 2 should be more informative about the limits between Himalaya, Hindu Kush and Karakoram

Lines 212-223: Authors should be more explicit on the significance of identified predicting parameters for ibex and blue sheep (e.g., what does exactly mean “precipitation seasonality”? According to analyses, which season should be the rainmost in a blue sheep suitable range?). I think a dedicated table is necessary and I’m sure that readers would appreciate

Lines 293-240: Please, add what does it mean in terms of % reduction of suitable surface

Lines 241-243: Table 2 is not clear. Do numbers refer to pixels? For the sake of clarity, please add an additional right column with “TOTAL”

Line 283: insert something like “although at the same time thinning” after “expanding”. This is very important to stress

Line 290: this information ….“based on current climatic data (1970-2000)”….should better appear under M&M

Line 291-292: “These predictions are supported by the existing literature” should be deleted (I suggest this option) or completed with citations

Lines 300-302: a bit redundant. I would delete the sentence, and start with something like “Both ibex and blue sheep habitats are usually ….”

Lines 302-318: as already signalled (see suggestion lines 212-223) Authors should describe more precisely the variables which contribute more in predicting unsuitable scenarios (eg, annual mean temperature above or below what?)

Line 321: add a citation number (not a year) after Aryal et al.

Line 326-327: “or towards a northeast direction” should be either deleted or supported by (at least) a citation

Line 329: “began” instead of “begin”

Lines 346: it would be desirable to have the boundaries of the mentioned NPs outlined on (at least) one of the figures

Line 358: “is likely due” sounds better

Lines 368-369: I suggest to delete “To do so …… predicted habitats”, since decisions on park boundaries are usually not driven by conservation needs related to one/two mammalian species enjoying a relatively favorable conservation status

6. PLOS authors have the option to publish the peer review history of their article (what does this mean?). If published, this will include your full peer review and any attached files.

Reviewer #1: No

---

## [Decision Letter · Decision Letter 1]

2 Nov 2021

Expanding or shrinking? range shifts in wild ungulates under climate change in Pamir-Karakoram Mountains, Pakistan

PONE-D-21-17916R1

Dear Dr. Nawaz,

We’re pleased to inform you that your manuscript has been judged scientifically suitable for publication and will be formally accepted for publication once it meets all outstanding technical requirements.

Kind regards,

Tzen-Yuh Chiang

Academic Editor

PLOS ONE

Additional Editor Comments (optional):

Reviewers' comments:

Reviewer's Responses to Questions

**Comments to the Author**

1. If the authors have adequately addressed your comments raised in a previous round of review and you feel that this manuscript is now acceptable for publication, you may indicate that here to bypass the “Comments to the Author” section, enter your conflict of interest statement in the “Confidential to Editor” section, and submit your "Accept" recommendation.

Reviewer #1: All comments have been addressed

2. Is the manuscript technically sound, and do the data support the conclusions?

Reviewer #1: Yes

3. Has the statistical analysis been performed appropriately and rigorously? 

Reviewer #1: Yes

4. Have the authors made all data underlying the findings in their manuscript fully available?

Reviewer #1: Yes

5. Is the manuscript presented in an intelligible fashion and written in standard English?

Reviewer #1: Yes

6. Review Comments to the Author

Reviewer #1: I'm fully satisfied of authors' reaction to my previous comments/suggestions. Some editing inaccuracies were still present (see attached file)

7. PLOS authors have the option to publish the peer review history of their article (what does this mean?). If published, this will include your full peer review and any attached files.

Reviewer #1: No

---

## [Editor Report · Acceptance letter]

22 Dec 2021

PONE-D-21-17916R1 

Expanding or shrinking? range shifts in wild ungulates under climate change in Pamir-Karakoram Mountains, Pakistan 

Dear Dr. Nawaz:

I'm pleased to inform you that your manuscript has been deemed suitable for publication in PLOS ONE. Congratulations! Your manuscript is now with our production department. 

Kind regards, 

on behalf of

Dr. Tzen-Yuh Chiang 

Academic Editor

PLOS ONE